# TRAINING WITH QUANTIZATION NOISE FOR EXTREME MODEL COMPRESSION

**Pierre Stock** [*†]
Facebook AI Research, Inria

**Angela Fan**[*]
Facebook AI Research, LORIA

**Benjamin Graham**
Facebook AI Research

**Edouard Grave**
Facebook AI Research

**Rémi Gribonval** [†]
Inria

**Hervé Jégou**
Facebook AI Research

**Armand Joulin**
Facebook AI Research

## ABSTRACT

We tackle the problem of producing compact models, maximizing their accuracy for a given model size. A standard solution is to train networks with Quantization Aware Training (Jacob et al., 2018), where the weights are quantized during training and the gradients approximated with the Straight-Through Estimator (Bengio et al., 2013). In this paper, we extend this approach to work beyond `int8` fixed-point quantization with extreme compression methods where the approximations introduced by STE are severe, such as Product Quantization. Our proposal is to only quantize a different random subset of weights during each forward, allowing for unbiased gradients to flow through the other weights. Controlling the amount of noise and its form allows for extreme compression rates while maintaining the performance of the original model. As a result we establish new state-of-the-art compromises between accuracy and model size both in natural language processing and image classification. For example, applying our method to state-of-the-art Transformer and ConvNet architectures, we can achieve 82.5% accuracy on MNLI by compressing RoBERTa to 14 MB and 80.0% top-1 accuracy on ImageNet by compressing an EfficientNet-B3 to 3.3 MB. [1]

## 1 INTRODUCTION

Many of the best performing neural network architectures in real-world applications have a large number of parameters. For example, the current standard machine translation architecture, Transformer (Vaswani et al., 2017), has layers that contain millions of parameters. Even models that are designed to jointly optimize the performance and the parameter efficiency, such as EfficientNets (Tan & Le, 2019), still require dozens to hundreds of megabytes, which limits their applications to domains like robotics or virtual assistants.

Model compression schemes reduce the memory footprint of overparametrized models. Pruning (LeCun et al., 1990) and distillation (Hinton et al., 2015) remove parameters by reducing the number of network weights. In contrast, quantization focuses on reducing the bits per weight. This makes quantization particularly interesting when compressing models that have already been carefully optimized in terms of network architecture. Whereas deleting weights or whole hidden units will inevitably lead to a drop in performance, we demonstrate that quantizing the weights can be performed with little to no loss in accuracy.

Popular postprocessing quantization methods, like scalar quantization, replace the floating-point weights of a trained network by a lower-precision representation, like fixed-width integers (Vanhoucke et al., 2011). These approaches achieve a good compression rate with the additional benefit of accelerating inference on supporting hardware. However, the errors made by these approximations

---

[*]Equal contribution. Corresponding authors: `pstock@fb.com`, `angelafan@fb.com`
[†]Univ Lyon, Inria, CNRS, ENS de Lyon, UCB Lyon 1, LIP UMR 5668, F-69342, Lyon, France
[1]Code available at `https://github.com/pytorch/fairseq/tree/master/examples/quant_noise`

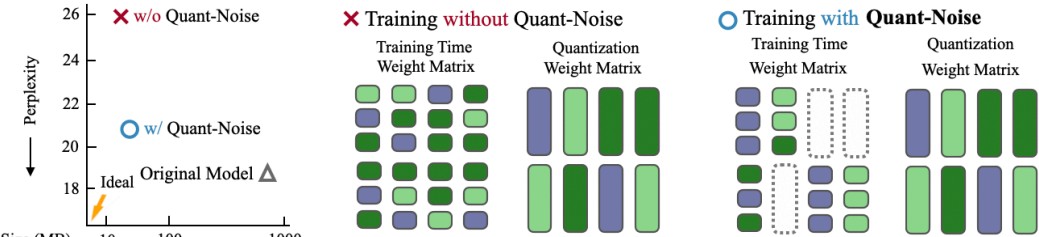

Figure 1: **Quant-Noise** trains models to be resilient to inference-time quantization by mimicking the effect of the quantization method during training time. This allows for extreme compression rates without much loss in accuracy on a variety of tasks and benchmarks.

accumulate in the computations operated during the forward pass, inducing a significant drop in performance (Stock et al., 2019).

A solution to address this drifting effect is to directly quantize the network during training. This raises two challenges. First, the discretization operators have a null gradient — the derivative with respect to the input is zero almost everywhere. This requires special workarounds to train a network with these operators. The second challenge that often comes with these workarounds is the discrepancy that appears between the train and test functions implemented by the network. Quantization Aware Training (QAT) (Jacob et al., 2018) resolves these issues by quantizing all the weights during the forward and using a straight through estimator (STE) (Bengio et al., 2013) to compute the gradient. This works when the error introduced by STE is small, like with `int8` quantization, but does not suffice in compression regimes where the approximation made by the compression is more severe.

In this work, we show that quantizing only a subset of weights instead of the entire network during training is more stable for high compression schemes. Indeed, by quantizing only a random fraction of the network at each forward, most the weights are updated with unbiased gradients. Interestingly, we show that our method can employ a simpler quantization scheme during the training. This is particularly useful for quantizers with trainable parameters, such as Product Quantizer (PQ), for which our quantization proxy is not parametrized. Our approach simply applies a quantization noise, called Quant-Noise, to a random subset of the weights, see Figure 1. We observe that this makes a network resilient to various types of discretization methods: it significantly improves the accuracy associated with (a) low precision representation of weights like `int8`; and (b) state-of-the-art PQ. Further, we demonstrate that Quant-Noise can be applied to existing trained networks as a post-processing step, to improve the performance network after quantization.

In summary, this paper makes the following contributions:

- We introduce the Quant-Noise technique to learn networks that are more resilient to a variety of quantization methods such as `int4`, `int8`, and PQ;
- Adding Quant-Noise to PQ leads to new state-of-the-art trade-offs between accuracy and model size. For instance, for natural language processing (NLP), we reach 82.5% accuracy on MNLI by compressing RoBERTa to 14 MB. Similarly for computer vision, we report 80.0% top-1 accuracy on ImageNet by compressing an EfficientNet-B3 to 3.3 MB;
- By combining PQ and `int8` to quantize weights and activations for networks trained with Quant-Noise, we obtain extreme compression with fixed-precision computation and achieve 79.8% top-1 accuracy on ImageNet and 21.1 perplexity on WikiText-103.

## 2 RELATED WORK

**Model compression.** Many compression methods focus on efficient parameterization, via weight pruning (LeCun et al., 1990; Li et al., 2016; Huang et al., 2018; Mittal et al., 2018), weight sharing (Dehghani et al., 2018; Turc et al., 2019; Lan et al., 2019) or with dedicated architectures (Tan & Le, 2019; Zhang et al., 2017; Howard et al., 2019). Weight pruning is implemented during training (Louizos et al., 2017) or as a fine-tuning post-processing step (Han et al., 2015; 2016). Many pruning methods are unstructured, i.e., remove individual weights (LeCun et al., 1990; Molchanov et al., 2017). On the other hand, structured pruning methods follow the structure of the weights to

reduce both the memory footprint and the inference time of a model (Li et al., 2016; Luo et al., 2017; Fan et al., 2019). We refer the reader to Liu et al. (2018) for a review of different pruning strategies.

Other authors have worked on lightweight architectures, by modifying existing models (Zhang et al., 2018; Wu et al., 2019; Sukhbaatar et al., 2019a) or developing new networks, such as MobileNet (Howard et al., 2019), ShuffleNet (Zhang et al., 2017), and EfficientNet (Tan & Le, 2019) in vision.

Finally, knowledge distillation (Hinton et al., 2015) has been applied to sentence representation (Turc et al., 2019; Sanh et al., 2019a; Sun et al., 2019; Zhao et al., 2019; Jiao et al., 2019), to reduce the size of a BERT model (Devlin et al., 2018).

**Quantization.** There are extensive studies of scalar quantization to train networks with low-precision weights and activations (Courbariaux et al., 2015; Courbariaux & Bengio, 2016; Rastegari et al., 2016; McDonnell, 2018). These methods benefit from specialized hardware to also improve the runtime during inference (Vanhoucke et al., 2011). Other quantization methods such as Vector Quantization (VQ) and PQ (Jegou et al., 2011) quantize blocks of weights simultaneously to achieve higher compression rate (Stock et al., 2019; Gong et al., 2014; Joulin et al., 2016; Carreira-Perpiñán & Idelbayev, 2017). Closer to our work, several works have focused at simultaneously training and quantizing a network (Jacob et al., 2018; Krishnamoorthi, 2018; Gupta et al., 2015; Dong et al., 2019). Gupta et al. (2015) assigns weights to a quantized bin stochastically which is specific to scalar quantization, but allows training with fixed point arithmetic. Finally, our method can be interpreted as a form of Bayesian compression (Louizos et al., 2017), using the Bayesian interpretation of Dropout (Gal & Ghahramani, 2016). As opposed to their work, we select our noise to match the weight transformation of a target quantization methods without restricting it to a scale mixture prior.

## 3 QUANTIZING NEURAL NETWORKS

In this section, we present the principles of quantization, several standard quantization methods, and describe how to combine scalar and product quantization. For clarity, we focus on the case of a fixed real matrix $\mathbf{W} \in \mathbf{R}^{n \times p}$. We suppose that this matrix is split into $m \times q$ blocks $\mathbf{b}_{kl}$:

$$\mathbf{W} = \begin{pmatrix} \mathbf{b}_{11} & \cdots & \mathbf{b}_{1q} \\ \vdots & \ddots & \vdots \\ \mathbf{b}_{m1} & \cdots & \mathbf{b}_{mq} \end{pmatrix}, \tag{1}$$

where the nature of these blocks is determined by the quantization method. A codebook is a set of $K$ vectors, i.e., $\mathcal{C} = \{\mathbf{c}[1], \ldots, \mathbf{c}[K]\}$. Quantization methods compress the matrix $\mathbf{W}$ by assigning to each block $\mathbf{b}_{kl}$ an index that points to a codeword $\mathbf{c}$ in a codebook $\mathcal{C}$, and storing the codebook $\mathcal{C}$ and the resulting indices (as the entries $\mathbf{I}_{kl}$ of an index matrix $\mathbf{I}$) instead of the real weights. During the inference, they reconstruct an approximation $\widehat{\mathbf{W}}$ of the original matrix $\mathbf{W}$ such that $\widehat{\mathbf{b}}_{kl} = \mathbf{c}[\mathbf{I}_{kl}]$.

We distinguish scalar quantization, such as int8, where each block $\mathbf{b}_{kl}$ consists of a single weight, from vector quantization, where several weights are quantized jointly.

### 3.1 FIXED-POINT SCALAR QUANTIZATION

Fixed-point scalar quantization methods replace floating-point number representations by low-precision fixed-point representations. They simultaneously reduce a model's memory footprint and accelerate inference by using fixed-point arithmetic on supporting hardware.

Fixed-point scalar quantization operates on blocks that represent a single weight, i.e., $\mathbf{b}_{kl} = \mathbf{W}_{kl}$. Floating-point weights are replaced by $N$ bit fixed-point numbers (Gupta et al., 2015), with the extreme case of binarization where $N = 1$ (Courbariaux et al., 2015). More precisely, the weights are rounded to one of $2^N$ possible codewords. These codewords correspond to bins evenly spaced by a scale factor $s$ and shifted by a bias $z$. Each weight $\mathbf{W}_{kl}$ is mapped to its nearest codeword $c$ by successively quantizing with $z \mapsto \text{round}(\mathbf{W}_{kl}/s + z)$ and dequantizing with the inverse operation:

$$\mathbf{c} = (\text{round}(\mathbf{W}_{kl}/s + z) - z) \times s, \tag{2}$$

where we compute the scale and bias as:

$$s = \frac{\max \mathbf{W} - \min \mathbf{W}}{2^N - 1} \quad \text{and} \quad z = \text{round}(\min \mathbf{W}/s).$$

We focus on this uniform rounding scheme instead of other non-uniform schemes (Choi et al., 2018; Li et al., 2019), because it allows for fixed-point arithmetic with implementations in PyTorch and Tensorflow (see Appendix). The compression rate is $\times 32/N$. The activations are also rounded to $N$-bit fixed-point numbers. With `int8` for instance, this leads to $\times 2$ to $\times 4$ faster inference on dedicated hardware. In this work, we consider both `int4` and `int8` quantization.

## 3.2 Product Quantization

Several quantization methods work on groups of weights, such as vectors, to benefit from the correlation induced by the structure of the network. In this work, we focus on Product Quantization for its good performance at extreme compression ratio (Stock et al., 2019).

**Traditional PQ.** In vector quantization methods, the blocks are predefined groups of weights instead of single weights. The codewords are groups of values, and the index matrix $\mathbf{I}$ maps groups of weights from the matrix $\mathbf{W}$ to these codewords. In this section, we present the Product Quantization framework as it generalizes both scalar and vector quantization. We consider the case where we apply PQ to the *columns* of $\mathbf{W}$ and thus assume that $q = p$.

Traditional vector quantization techniques split the matrix $\mathbf{W}$ into its $p$ columns and learn a codebook on the resulting $p$ vectors. Instead, Product Quantization splits each column into $m$ subvectors and learns the same codebook for each of the resulting $m \times p$ subvectors. Each quantized vector is subsequently obtained by assigning its subvectors to the nearest codeword in the codebook. Learning the codebook is traditionally done using $k$-means with a fixed number $K$ of centroids, typically $K = 256$ to store the index matrix $\mathbf{I}$ using `int8`. Thus, the objective function is written as:

$$\|\mathbf{W} - \widehat{\mathbf{W}}\|_2^2 = \sum_{k,l} \|\mathbf{b}_{kl} - \mathbf{c}[\mathbf{I}_{kl}]\|_2^2. \tag{3}$$

PQ shares representations between subvectors, which allows for higher compression rates than `intN`.

**Iterative PQ.** When quantizing a full network rather than a single matrix, extreme compression with PQ induces a quantization drift as reconstruction error accumulates (Stock et al., 2019). Indeed, subsequent layers take as input the output of preceding layers, which are modified by the quantization of the preceding layers. This creates a drift in the network activations, resulting in large losses of performance. A solution proposed by Stock et al. (2019), which we call **iterative PQ** (iPQ), is to quantize layers sequentially from the lowest to the highest, and finetune the upper layers as the lower layers are quantized, under the supervision of the uncompressed (teacher) model. Codewords of each layer are finetuned by averaging the gradients of their assigned elements with gradient steps:

$$\mathbf{c} \leftarrow \mathbf{c} - \eta \frac{1}{|J_{\mathbf{c}}|} \sum_{(k,l) \in J_{\mathbf{c}}} \frac{\partial \mathcal{L}}{\partial \mathbf{b}_{kl}}, \tag{4}$$

where $J_{\mathbf{c}} = \{(k,l) \mid \mathbf{c}[\mathbf{I}_{kl}] = \mathbf{c}\}$, $\mathcal{L}$ is the loss function and $\eta > 0$ is a learning rate. This adapts the upper layers to the drift appearing in their inputs, reducing the impact of the quantization approximation on the overall performance.

## 3.3 Combining Fixed-Point with Product Quantization

Fixed-point quantization and Product Quantization are often regarded as competing choices, but can be advantageously combined. Indeed, PQ/iPQ compresses the network by replacing vectors of weights by their assigned centroids, but these centroids are in floating-point precision. Fixed-point quantization compresses both activations and weights to fixed-point representations. Combining both approaches means that the vectors of weights are mapped to centroids that are compressed to fixed-point representations, along with the activations. This benefits from the extreme compression ratio of iPQ and the finite-precision arithmetics of `intN` quantization.

More precisely, for a given matrix, we store the `int8` representation of the $K$ centroids of dimension $d$ along with the $\log_2 K$ representations of the centroid assignments of the $m \times p$ subvectors. The `int8` representation of the centroids is obtained with Eq. (2). The overall storage of the matrix and activations during a forward pass with batch size 1 (recalling that the input dimension is n) writes

$$M = 8 \times Kd + \log_2 K \times mp + 8 \times n \text{ bits.} \tag{5}$$

In particular, when $K = 256$, the centroid assignments are also stored in `int8`, which means that every value required for a forward pass is stored in an `int8` format. We divide by 4 the `float32` overhead of storing the centroids, although the storage requirement associated with the centroids is small compared to the cost of indexing the subvectors for standard networks. In contrast to iPQ alone where we only quantize the weights, we also quantize the activations using `int8`. We evaluate this approach on both natural language processing and computer vision tasks in Section 5.

## 4 METHOD

Deep networks are not exposed to the noise caused by the quantization drift during training, leading to suboptimal performance. A solution to make the network robust to quantization is to introduce it during training. Quantization Aware Training (QAT) (Jacob et al., 2018) exposes the network during training by quantizing weights during the forward pass. This transformation is not differentiable and gradients are approximated with a straight through estimator (STE) (Bengio et al., 2013; Courbariaux & Bengio, 2016). STE introduces a bias in the gradients that depends on level of quantization of the weights, and thus, the compression ratio. In this section, we propose a simple modification to control this induced bias with a stochastic amelioration of QAT, called Quant-Noise. The idea is to quantize a randomly selected fraction of the weights instead of the full network as in QAT, leaving some unbiased gradients flow through unquantized weights. Our general formulation can simulate the effect of both quantization and of pruning during training.

### 4.1 TRAINING NETWORKS WITH QUANTIZATION NOISE

We consider the case of a real matrix $\mathbf{W}$ as in Section 3. During the training of a network, our proposed Quant-Noise method works as follows: first, we compute blocks $\mathbf{b}_{kl}$ related to a target quantization method. Then, during each forward pass, we randomly select a subset of these blocks and apply some distortion to them.

More formally, given a set of tuples of indices $J \subset \{(k, l)\}$ for $1 \leq k \leq m$, $1 \leq l \leq q$ and a *distortion* or *noise* function $\varphi$ acting on a block, we define an operator $\psi(\cdot \mid J)$ such that, for each block $\mathbf{b}_{kl}$, we apply the following transformation:

$$\psi(\mathbf{b}_{kl} \mid J) = \begin{cases} \varphi(\mathbf{b}_{kl}) & \text{if } (k, l) \in J, \\ \mathbf{b}_{kl} & \text{otherwise.} \end{cases} \tag{6}$$

The noise function $\varphi$ simulates the change in the weights produced by the target quantization method (see Section 4.2 for details). We replace the matrix $\mathbf{W}$ by the resulting noisy matrix $\mathbf{W}_{\text{noise}}$ during the forward pass to compute a noisy output $\mathbf{y}_{\text{noise}}$, i.e.,

$$\mathbf{W}_{\text{noise}} = (\psi(\mathbf{b}_{kl} \mid J))_{kl} \quad \text{and} \quad \mathbf{y}_{\text{noise}} = \mathbf{x}\mathbf{W}_{\text{noise}} \tag{7}$$

where $\mathbf{x}$ is an input vector. During the backward pass, we apply STE, which amounts to replacing the distorted weights $\mathbf{W}_{\text{noise}}$ by their non-distorted counterparts. Note that our approach is equivalent to QAT when $J$ contaits all the tuples of indices. However, an advantage of Quant-Noise over QAT is that unbiased gradients continue to flow via blocks unaffected by the noise. As these blocks are randomly selected for each forward, we guarantee that each weight regularly sees gradients that are not affected by the nature of the function $\varphi$. As a side effect, our quantization noise regularizes the network in a similar way as DropConnect (Wan et al., 2013) or LayerDrop (Fan et al., 2019).

**Composing quantization noises.** As noise operators are compositionally commutative, we can make a network robust to a combination of quantization methods by composing their noise operators:

$$\psi(\mathbf{b}_{kl} \mid J) = \psi_1 \circ \psi_2(\mathbf{b}_{kl} \mid J). \tag{8}$$

This property is particularly useful to combine quantization with pruning operators during training, as well as combining scalar quantization with product quantization.

### 4.2 ADDING NOISE TO SPECIFIC QUANTIZATION METHODS

In this section, we propose several implementations of the noise function $\varphi$ for the quantization methods described in Section 3. We also show how to handle pruning with it.

| Quantization Scheme | Language Modeling 16-layer Transformer Wikitext-103 | | | Image Classification EfficientNet-B3 ImageNet-1k | | |
|---|---|---|---|---|---|---|
| | Size | Compression | PPL | Size | Compression | Top-1 |
| Uncompressed model | 942 | × 1 | 18.3 | 46.7 | × 1 | 81.5 |
| `int4` quantization | 118 | × 8 | 39.4 | 5.8 | × 8 | 45.3 |
| - trained with QAT | 118 | × 8 | 34.1 | 5.8 | × 8 | 59.4 |
| - trained with Quant-Noise | 118 | × 8 | **21.8** | 5.8 | × 8 | **67.8** |
| `int8` quantization | 236 | × 4 | 19.6 | 11.7 | × 4 | 80.7 |
| - trained with QAT | 236 | × 4 | 21.0 | 11.7 | × 4 | 80.8 |
| - trained with Quant-Noise | 236 | × 4 | **18.7** | 11.7 | × 4 | **80.9** |
| iPQ | 38 | × 25 | 25.2 | 3.3 | × 14 | 79.0 |
| - trained with QAT | 38 | × 25 | 41.2 | 3.3 | × 14 | 55.7 |
| - trained with Quant-Noise | 38 | × 25 | **20.7** | 3.3 | × 14 | **80.0** |
| iPQ & `int8` + Quant-Noise | 38 | × 25 | 21.1 | 3.1 | × 15 | 79.8 |

Table 1: **Comparison of different quantization schemes with and without Quant-Noise** on language modeling and image classification. For language modeling, we train a Transformer on the Wikitext-103 benchmark and report perplexity (PPL) on test. For image classification, we train a EfficientNet-B3 on the ImageNet-1k benchmark and report top-1 accuracy on validation and use our re-implementation of EfficientNet-B3. The original implementation of Tan & Le (2019) achieves an uncompressed Top-1 accuracy of $81.9\%$. For both settings, we report model size in megabyte (MB) and the compression ratio compared to the original model.

**Fixed-point scalar quantization.** In `intN` quantization, the blocks are atomic and weights are rounded to their nearest neighbor in the codebook. The function $\varphi$ replaces weight $\mathbf{W}_{kl}$ with the output of the rounding function defined in Eq. (2), i.e.,

$$\varphi_{\texttt{intN}}(w) = (\text{round}(w/s + z) - z) \times s, \tag{9}$$

where $s$ and $z$ are updated during training. In particular, the application of Quant-Noise to `int8` scalar quantization is a stochastic amelioration of QAT.

**Product quantization.** As opposed to `intN`, codebooks in PQ require a clustering step based on weight values. During training, we learn codewords online and use the resulting centroids to implement the quantization noise. More precisely, the noise function $\varphi_{\text{PQ}}$ assigns a selected block $\mathbf{b}$ to its nearest codeword in the associated codebook $\mathcal{C}$:

$$\varphi_{\text{PQ}}(\mathbf{v}) = \text{argmin}_{\mathbf{c} \in \mathcal{C}} \|\mathbf{b} - \mathbf{c}\|_2^2. \tag{10}$$

Updating the codebooks online works well. However, empirically, running $k$-means once per epoch is faster and does not noticeably modify the resulting accuracy.

Note that computing the exact noise function for PQ is computationally demanding. We propose a simpler and faster alternative approximation $\varphi_{\text{proxy}}$ to the operational transformation of PQ and iPQ. The noise function simply zeroes out the subvectors of the selected blocks, i.e., $\varphi_{\text{proxy}}(\mathbf{v}) = 0$. As a sidenote, we considered other alternatives, for instance one where the subvectors are mapped to the mean subvector. In practice, we found that these approximations lead to similar performance, see Section 7.2. This proxy noise function is a form of Structured Dropout and encourages correlations between the subvectors. This correlation is beneficial to the subsequent clustering involved in PQ/iPQ.

**Adding pruning to the quantization noise.** The specific form of quantization noise can be adjusted to incorporate additional noise specific to pruning. We simply combine the noise operators of quantization and pruning by composing them following Eq. (8). We consider the pruning noise function of Fan et al. (2019) where they randomly drop predefined structures during training. In particular, we focus on *LayerDrop*, where the structures are the residual blocks of highway-like layers (Srivastava et al., 2015), as most modern architectures, such as ResNet or Transformer, are composed of this structure. More precisely, the corresponding noise operator over residual blocks $\mathbf{v}$ is $\varphi_{\text{LayerDrop}}(\mathbf{v}) = 0$. For pruning, we do not use STE to backpropagate the gradient of pruned weights, as dropping them entirely during training has the benefit of speeding convergence (Huang

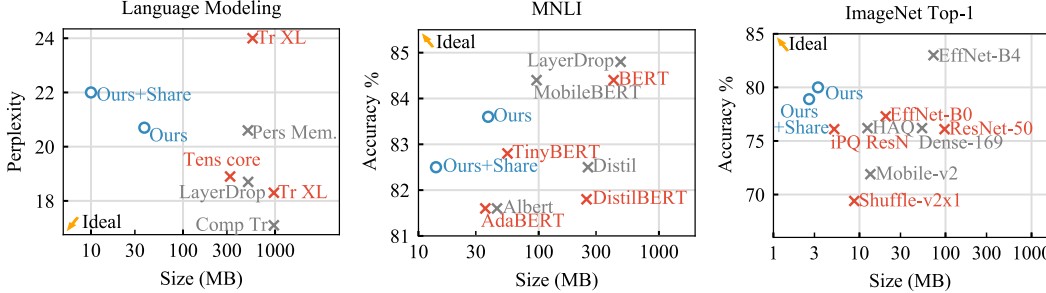

Figure 2: **Performance as a function of model size.** We compare models quantized with PQ and trained with the related Quant-Noise to the state of the art. **(a)** Test perplexity on Wikitext-103 **(b)** Dev Accuracy on MNLI **(c)** ImageNet Top-1 accuracy. Model size is shown in megabytes on a log scale. Red and gray coloring indicates existing work, with different colors for visual distinction.

| | Language modeling | | | Sentence Representation | | | Image Classification | | |
|---|---|---|---|---|---|---|---|---|---|
| | Comp. | Size | PPL | Comp. | Size | Acc. | Comp. | Size | Acc. |
| *Unquantized models* | | | | | | | | | |
| Original model | × 1 | 942 | 18.3 | × 1 | 480 | 84.8 | × 1 | 46.7 | 81.5 |
| + Sharing | × 1.8 | 510 | 18.7 | × 1.9 | 250 | 84.0 | × 1.4 | 34.2 | 80.1 |
| + Pruning | × 3.7 | 255 | 22.5 | × 3.8 | 125 | 81.3 | × 1.6 | 29.5 | 78.5 |
| *Quantized models* | | | | | | | | | |
| iPQ | × 24.8 | 38 | 25.2 | × 12.6 | 38 | 82.5 | × 14.1 | 3.3 | 79.0 |
| + Quant-Noise | × 24.8 | 38 | 20.7 | × 12.6 | 38 | 83.6 | × 14.1 | 3.3 | 80.0 |
| + Sharing | × 49.5 | 19 | 22.0 | × 34.3 | 14 | 82.5 | × 18 | 2.6 | 78.9 |
| + Pruning | × 94.2 | 10 | 24.7 | × 58.5 | 8 | 78.8 | × 20 | 2.3 | 77.8 |

Table 2: **Decomposing the impact of the different compression schemes.** **(a)** we train Transformers with Adaptive Input and LayerDrop on Wikitext-103 **(b)** we pre-train RoBERTA base models with LayerDrop and then finetune on MNLI **(c)** we train an EfficientNet-B3 on ImageNet. We report the compression ratio w.r.t. to the original model ("comp.") and the resulting size in MB.

et al., 2016). Once a model is trained with LayerDrop, the number of layers kept at inference can be adapted to match computation budget or time constraint.

## 5 RESULTS

We demonstrate the impact of Quant-Noise on the performance of several quantization schemes in a variety of settings (see Appendix - Sec. 7.5).

### 5.1 IMPROVING COMPRESSION WITH QUANT-NOISE

Quant-Noise is a regularization method that makes networks more robust to the target quantization scheme or combination of quantization schemes during training. We show the impact of Quant-Noise in Table 1 for a variety of quantization methods: `int8`/`int4` and iPQ.

We experiment in 2 different settings: a Transformer network trained for language modeling on WikiText-103 and a EfficientNet-B3 convolutional network trained for image classification on ImageNet-1k. Our quantization noise framework is general and flexible — Quant-Noise improves the performance of quantized models for every quantization scheme in both experimental settings. Importantly, Quant-Noise only changes model training by adding a regularization noise similar to dropout, with no impact on convergence and very limited impact on training speed (< 5% slower).

This comparison of different quantization schemes shows that Quant-Noise works particularly well with high performance quantization methods, like iPQ, where QAT tends to degrade the performances, even compared to quantizing as a post-processing step. In subsequent experiments in this section, we focus on applications with iPQ because it offers the best trade-off between model performance and compression, and has little negative impact on FLOPS.

| Language Modeling | PPL | RoBERTa | Acc. |
|---|---|---|---|
| Train without Quant-Noise | 25.2 | Train without Quant-Noise | 82.5 |
| + Finetune with Quant-Noise | 20.9 | + Finetune with Quant-Noise | 83.4 |
| Train with Quant-Noise | 20.7 | Train with Quant-Noise | 83.6 |

Table 3: **Quant-Noise: Finetuning vs training.** We report performance after iPQ quantization. We train with the $\phi_{\text{proxy}}$ noise and finetune with Quant-Noise, and use it during the transfer to MNLI for each RoBERTa model.

**Fixed-Point Product Quantization.** Combining iPQ and `int8` as described in Section 3.3 allows us to take advantage of the high compression rate of iPQ with a fixed-point representation of both centroids and activations. As shown in Table 1, this combination incurs little loss in accuracy with respect to iPQ + Quant-Noise. Most of the memory footprint of iPQ comes from indexing and not storing centroids, so the compression ratios are comparable.

**Complementarity with Weight Pruning and Sharing.** We analyze how Quant-Noise is compatible and complementary with pruning ("+Prune") and weight sharing ("+Share"), see Appendix for details on weight sharing. We report results for Language modeling on WikiText-103, pre-trained sentence representations on MNLI and object classification on ImageNet-1k in Table 2. The conclusions are remarkably consistent across tasks and benchmarks: Quant-Noise gives a large improvement over strong iPQ baselines. Combining it with sharing and pruning offers additional interesting operating points of performance vs size.

## 5.2 COMPARISON WITH THE STATE OF THE ART

We now compare our approach on the same tasks against the state of the art. We compare iPQ + Quant-Noise with 6 methods of network compression for Language modeling, 8 state-of-the-art methods for Text classification, and 8 recent methods evaluate image classification on Imagenet with compressed models. These comparisons demonstrate that Quant-Noise leads to extreme compression rates at a reasonable cost in accuracy. We apply our best quantization setup on competitive models and reduce their memory footprint by $\times 20 - 94$ when combining with weight sharing and pruning, offering extreme compression for good performance.

**Natural Language Processing.** In Figure 2, we examine the trade-off between performance and model size. Our quantized RoBERTa offers a competitive trade-off between size and performance with memory reduction methods dedicated to BERT, like TinyBERT, MobileBERT, or AdaBERT.

**Image Classification.** We compress EfficientNet-B3 from $46.7$Mb to $3.3$Mb ($\times 14$ compression) while maintaining high top-1 accuracy ($78.5\%$ versus $80\%$ for the original model). As shown in Figure 2, our quantized EfficientNet-B3 is smaller and more accurate than architectures dedicated to optimize on-device performance with limited size like MobileNet or ShuffleNet. We further evaluate the beneficial effect of Quant-Noise on ResNet-50 to compare directly with Stock et al. (2019). Results shown in Table 4 indicate improvement with Quant-Noise compared to previous work.

Incorporating pruning noise into quantization is also beneficial. For example, with pruning iPQ+Quant-Noise reduces size by $\times 25$ with only a drop of $2.4$ PPL in language modeling. Further, pruning reduces FLOPS by the same ratio as its compression factor, in our case, $\times 2$. By adding sharing with pruning, in language modeling, we achieve an extreme compression ratio of $\times 94$ with a drop of $6.4$ PPL with FLOPS reduction from pruning entire shared chunks of layers. For comparison, our 10 MB model has the same performance as the 570 MB Transformer-XL base.

## 5.3 FINETUNING WITH QUANT-NOISE FOR POST-PROCESSING QUANTIZATION

We explore taking existing models and post-processing with Quant-Noise instead of training from scratch. For language modeling, we train for 10 additional epochs. For RoBERTa, we train for 25k additional updates. Finetuning with Quant-Noise incorporates the benefits and almost matches training from scratch (Table 3). In language modeling, there is only a $0.2$ PPL difference. We further examine how to incorporate Quant-Noise more flexibly into pretraining RoBERTa. We take an already

trained RoBERTa model and incorporate Quant-Noise during sentence classification finetuning. This is effective at compressing while retaining accuracy after quantization.

## 6 CONCLUSION

We show that quantizing a random subset of weights during training maintains performance in the high quantization regime. We validate that Quant-Noise works with a variety of different quantization schemes on several applications in text and vision. Our method can be applied to a combination of iPQ and `int8` to benefit from extreme compression ratio and fixed-point arithmetic. Finally, we show that Quant-Noise can be used as a post-processing step to prepare already trained networks for subsequent quantization, to improve the performance of the compressed model.

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

| Setting | Model | Compression | Top-1 Accuracy |
|---------|-------|-------------|----------------|
| Small Blocks | Stock et al. (2019) | 19x | 73.8 |
| | Quant-Noise | 19x | **74.3** |
| Large Blocks | Stock et al. (2019) | 32x | 68.2 |
| | Quant-Noise | 32x | **68.8** |

Table 4: **Compression of ResNet-50 with Quant-Noise**. We compare to Stock et al. (2019) in both the small and large blocks regime. For fair comparison, we hold the compression rate constant. Quant-Noise provides improved performance in both settings.

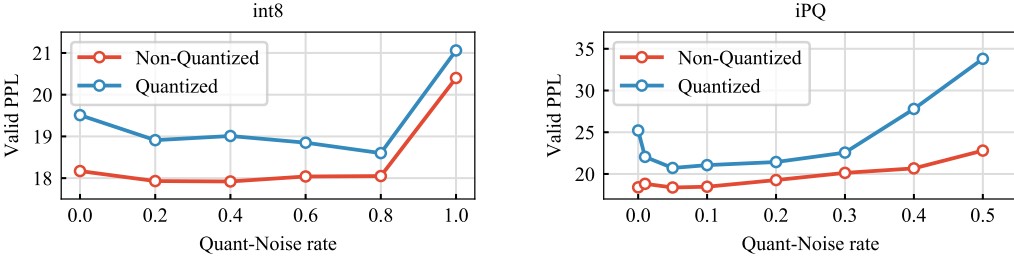

Figure 3: **Effect of Quantization Parameters**. We report the influence of the proportion of blocks to which we apply the noise. We focus on Transformer for Wikitext-103 language modeling. We explore two settings: iPQ and `int8`. For iPQ, we use $\varphi_{\text{proxy}}$.

# 7 APPENDIX

## 7.1 QUANTIZATION OF ADDITIONAL ARCHITECTURES

**ResNet-50.** We explore the compression of ResNet-50, a standard architecture used Computer Vision. In Table 4, we compare Quant-Noise to iPQ Compression from Stock et al. (2019) and show that Quant-Noise provide consistent additional improvement.

## 7.2 ABLATIONS

In this section, we examine the impact of the level of noise during training as well as the impact of approximating iPQ during training.

## 7.3 IMPACT OF NOISE RATE

We analyze the performance for various values of Quant-Noise in Figure 3 on a Transformer for language modeling. For iPQ, performance is impacted by high rates of quantization noise. For example, a Transformer with the noise function $\varphi_{\text{proxy}}$ degrades with rate higher than 0.5, i.e., when half of the weights are passed through the noise function $\varphi_{\text{proxy}}$. We hypothesize that for large quantities of noise, a larger effect of using proxy rather than the exact PQ noise is observed. For `int8` quantization and its noise function, higher rates of noise are slightly worse but not as severe. A rate of 1 for `int8` quantization is equivalent to the Quantization Aware Training of (Krishnamoorthi, 2018), as the full matrix is quantized with STE, showing the potential benefit of partial quantization during training.

## 7.4 IMPACT OF APPROXIMATING THE NOISE FUNCTION

We study the impact of approximating quantization noise during training. We focus on the case of iPQ with the approximation described in Section 4.2. In Table 5, we compare the correct noise function for iPQ with its approximation $\varphi_{\text{proxy}}$. This approximate noise function does not consider cluster assignments or centroid values and simply zeroes out the selected blocks. For completeness, we include an intermediate approximation where we consider cluster assignments to apply noise

| Noise | Blocks | PPL | Quant PPL |
|---|---|---|---|
| $\varphi_{PQ}$ | Subvectors | 18.3 | 21.1 |
| $\varphi_{PQ}$ | Clusters | 18.3 | 21.2 |
| $\varphi_{proxy}$ | Subvectors | 18.3 | 21.0 |
| $\varphi_{proxy}$ | Clusters | 18.4 | 21.1 |

Table 5: **Exact versus proxy noise function for different block selections with iPQ.** We compare exact $\phi_{PQ}$ and the approximation $\phi_{proxy}$ with blocks selected from all subvectors or subvectors from the same cluster.

within each cluster, but still zero-out the vectors. These approximations do not affect the performance of the quantized models. This suggests that increasing the correlation between subvectors that are jointly clustered is enough to maintain the performance of a model quantized with iPQ. Since PQ tends to work well on highly correlated vectors, such as activations in convolutional networks, this is not surprising. Using the approximation $\varphi_{proxy}$ presents the advantage of speed and practicality. Indeed, one does not need to compute cluster assignments and centroids for every layer in the network after each epoch. Moreover, the approach $\varphi_{proxy}$ is less involved in terms of code.

## 7.5 EXPERIMENTAL SETTING

We assess the effectiveness of Quant-Noise on competitive language and vision benchmarks. We consider Transformers for language modeling, RoBERTa for pre-training sentence representations, and EfficientNet for image classification. Our models are implemented in PyTorch (Paszke et al., 2017). We use `fairseq` (Ott et al., 2019) for language modeling and pre-training for sentence representation tasks and `Classy Vision` (Adcock et al., 2019) for EfficientNet.

**Language Modeling.** We experiment on the `Wikitext-103` benchmark (Merity et al., 2016) that contains 100M tokens and a vocabulary of 260k words. We train a 16 layer Transformer following Baevski & Auli (2018) with a LayerDrop rate of 0.2 (Fan et al., 2019). We report perplexity (PPL) on the test set.

**Pre-Training of Sentence Representations.** We pre-train the base BERT model (Devlin et al., 2018) on the `BooksCorpus + Wiki` dataset with a LayerDrop rate of 0.2. We finetune the pre-trained models on the MNLI task (Williams et al., 2018) from the GLUE Benchmark (Wang et al., 2019) and report accuracy. We follow the parameters in Liu et al. (2019) training and finetuning.

**Image Classification.** We train an EfficientNet-B3 model (Tan & Le, 2019) on the ImageNet object classification benchmark (Deng et al., 2009). The EfficientNet-B3 of `Classy Vision` achieves a Top-1 accuracy of 81.5%, which is slightly below than the performance of 81.9% reported by Tan & Le (2019).

## 7.6 TRAINING DETAILS

**Language Modeling** To handle the large vocabulary of Wikitext-103, we follow (Dauphin et al., 2017) and (Baevski & Auli, 2018) in using adaptive softmax (Grave et al., 2016) and adaptive input for computational efficiency. For both input and output embeddings, we use dimension size 1024 and three adaptive bands: 20K, 40K, and 200K. We use a cosine learning rate schedule (Baevski & Auli, 2018; Loshchilov & Hutter, 2016) and train with Nesterov's accelerated gradient (Sutskever et al., 2013). We set the momentum to 0.99 and renormalize gradients if the norm exceeds 0.1 (Pascanu et al., 2014). During training, we partition the data into blocks of contiguous tokens that ignore document boundaries. At test time, we respect sentence boundaries. We set LayerDrop to 0.2. We set Quant-Noise value to 0.05. During training time, we searched over the parameters (0.05, 0.1, 0.2) to determine the optimal value of Quant-Noise. During training time, the block size of Quant-Noise is 8.

**RoBERTa** The base architecture is a 12 layer model with embedding size 768 and FFN size 3072. We follow (Liu et al., 2019) in using the subword tokenization scheme from (Radford et al., 2019), which uses bytes as subword units. This eliminates unknown tokens. We train with large batches of size 8192 and maintain this batch size using gradient accumulation. We do not use next sentence prediction (Lample & Conneau, 2019). We optimize with Adam with a polynomial decay learning rate schedule. We set LayerDrop to 0.2. We set Quant-Noise value to 0.1. We did not hyperparameter

| Model | MB | PPL |
|---|---|---|
| Trans XL Large (Dai et al., 2019) | 970 | 18.3 |
| Compressive Trans (Rae et al., 2019) | 970 | 17.1 |
| GCNN (Dauphin et al., 2017) | 870 | 37.2 |
| 4 Layer QRNN (Bradbury et al., 2016) | 575 | 33.0 |
| Trans XL Base (Dai et al., 2019) | 570 | 24.0 |
| Persis Mem (Sukhbaatar et al., 2019b) | 506 | 20.6 |
| Tensorized core-2 (Ma et al., 2019) | 325 | 18.9 |
| Quant-Noise | **38** | 20.7 |
| Quant-Noise + Share + Prune | 10 | 24.2 |

Table 6: **Performance on Wikitext-103.** We report test set perplexity and model size in megabytes. Lower perplexity is better.

search to determine the optimal value of Quant-Noise as training RoBERTa is computationally intensive. During training time, the block size of Quant-Noise is 8.

During finetuning, we hyperparameter search over three learning rate options (1e-5, 2e-5, 3e-5) and batchsize (16 or 32 sentences). The other parameters are set following (Liu et al., 2019). We do single task finetuning, meaning we only tune on the data provided for the given natural language understanding task. We do not perform ensembling. When finetuning models trained with LayerDrop, we apply LayerDrop and Quant-Noise during finetuning time as well.

**EfficientNet** We use the architecture of EfficientNet-B3 defined in `Classy Vision` (Adcock et al., 2019) and follow the default hyperparameters for training. We set Quant-Noise value to 0.1. During training time, we searched over the parameters (0.05, 0.1, 0.2) to determine the optimal value of Quant-Noise. During training time, the block size of Quant-Noise is set to 4 for all $1 \times 1$ convolutions, 9 for depth-wise $3 \times 3$ convolutions, 5 for depth-wise $5 \times 5$ convolutions and 4 for the classifier. For sharing, we shared weights between blocks 9-10, 11-12, 14-15, 16-17, 19-20-21, 22-23 and refer to blocks that share the same weights as a *chunk*. For LayerDrop, we drop the chunks of blocks defined previously with probability 0.2 and evaluate only with chunks 9-10, 14-15 and 19-20-21.

### 7.7 SCALAR QUANTIZATION DETAILS

We closely follow the methodology of PyTorch 1.4. We emulate scalar quantization by quantizing the weights and the activations. The scales and zero points of activations are determined by doing a few forward passes ahead of the evaluation and then fixed. We use the `Histogram` method to compute $s$ and $z$, which aims at approximately minimizing the $L_2$ quantization error by adjusting $s$ and $z$. This scheme is a refinement of the `MinMax` scheme. Per channel quantization is also discussed in Table 10.

### 7.8 iPQ QUANTIZATION DETAILS

**Language Modeling** We quantize FFN with block size 8, embeddings with block size 8, and attention with block size 4. We tuned the block size for attention between the values (4, 8) to find the best performance. Note that during training with apply Quant-Noise to all the layers.

**RoBERTa** We quantize FFN with block size 4, embeddings with block size 4, and attention with block size 4. We tuned the block size between the values (4, 8) to find the best performance. Note that during training with apply Quant-Noise to all the layers.

**EfficientNet** We quantize blocks sequentially and end up with the classifier. The block sizes are 4 for all $1 \times 1$ convolutions, 9 for depth-wise $3 \times 3$ convolutions, 5 for depth-wise $5 \times 5$ convolutions and 4 for the classifier. Note that during training with apply Quant-Noise to all the weights in InvertedResidual Blocks (except the Squeeze-Excitation subblocks), the head convolution and the classifier.

| Model | MB | MNLI |
|---|---|---|
| RoBERTa Base + LD (Fan et al., 2019) | 480 | 84.8 |
| BERT Base (Devlin et al., 2018) | 420 | 84.4 |
| PreTrained Distil (Turc et al., 2019) | 257 | 82.5 |
| DistilBERT (Sanh et al., 2019b) | 250 | 81.8 |
| MobileBERT* (Sun et al.) | 96 | 84.4 |
| TinyBERT† (Jiao et al., 2019) | 55 | 82.8 |
| ALBERT Base (Lan et al., 2019) | 45 | 81.6 |
| AdaBERT† (Chen et al., 2020) | 36 | 81.6 |
| Quant-Noise | 38 | 83.6 |
| Quant-Noise + Share + Prune | 14 | 82.5 |

Table 7: **Performance on MNLI.** We report accuracy and size in megabytes. * indicates distillation using BERT Large. † indicates training with data augmentation. Work from Sun et al. (2019) and Zhao et al. (2019) do not report results on the dev set. Cao et al. do not report model size. Higher accuracy is better.

| Model | MB | Acc. |
|---|---|---|
| EfficientNet-B7 (Tan & Le, 2019) | 260 | 84.4 |
| ResNet-50 (He et al., 2015) | 97.5 | 76.1 |
| DenseNet-169 (Huang et al., 2018) | 53.4 | 76.2 |
| EfficientNet-B0 (Tan & Le, 2019) | 20.2 | 77.3 |
| MobileNet-v2 (Sandler et al., 2018) | 13.4 | 71.9 |
| Shufflenet-v2 ×1 (Ma et al., 2018) | 8.7 | 69.4 |
| HAQ 4 bits (Wang et al., 2018) | 12.4 | 76.2 |
| iPQ ResNet-50 (Stock et al., 2019) | 5.09 | 76.1 |
| Quant-Noise | 3.3 | 80.0 |
| Quant-Noise + Share + Prune | 2.3 | 77.8 |

Table 8: **Performance on ImageNet.** We report accuracy and size in megabytes. Higher accuracy is better.

## 7.9 DETAILS OF PRUNING AND LAYER SHARING

We apply the *Every Other Layer* strategy from Fan et al. (2019). When combining layer sharing with pruning, we train models with shared layers and then prune chunks of shared layers. When sharing layers, the weights of adjacent layers are shared in chunks of two. For a concrete example, imagine we have a model with layers A, B, C, D, E, F, G, H. We share layers A and B, C and D, E and F, G and H. To prune, every other chunk would be pruned away, for example we could prune A, B, E, F.

## 7.10 NUMERICAL RESULTS FOR GRAPHICAL DIAGRAMS

We report the numerical values displayed in Figures 2 in Table 6 for language modeling, Table 7 for BERT, and Table 8 for ImageNet.

## 7.11 FURTHER ABLATIONS

### 7.11.1 IMPACT OF QUANT-NOISE FOR THE VISION SETUP

We provide another study showing the impact of the proportion of elements on which to apply Quant-Noise in Table 9.

### 7.11.2 IMPACT OF THE NUMBER OF CENTROIDS

We quantize with 256 centroids which represents a balance between size and representation capacity. The effect of the number of centroids on performance and size is shown in Figure 4 (a). Quantizing

| $p$ | 0 | 0.2 | 0.4 | 0.6 | 0.8 | 1 |
|---|---|---|---|---|---|---|
| Top-1 | 80.66 | 80.83 | 80.82 | 80.88 | 80.92 | 80.64 |

Table 9: **Effect of Quantization Parameters**. We report the influence of the Quant-Noise rate $p$ with Scalar Quantization (`int8`). We focus on EfficientNet for ImageNet classification.

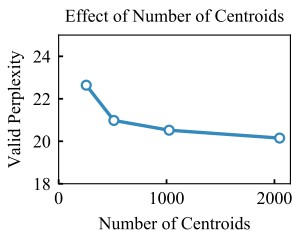

Figure 4: **Quantizing with a larger number of centroids**. Results are shown on Wikitext-103 valid.

with more centroids improves perplexity — this parameter could be adjusted based on the practical storage constraints.

### 7.11.3 EFFECT OF INITIAL MODEL SIZE

Large, overparameterized models are more easily compressed. In Figure 5, we explore quantizing both shallower and skinnier models. For shallow models, the gap between quantized and non-quantized perplexity does not increase as layers are removed (Figure 5, left). In contrast, there is a larger gap in performance for models with smaller FFN (Figure 5, right). As the FFN size decreases, the weights are less redundant and more difficult to quantize with iPQ.

### 7.11.4 DIFFICULTY OF QUANTIZING DIFFERENT MODEL STRUCTURES

Quantization is applied to various portions of the Transformer architecture — the embedding, attention, feedforward, and classifier output. We compare the quantizability of various portions of the network in this section.

**Is the order of structures important?** We quantize specific network structures first — this is important as quantizing weight matrices can accumulate reconstruction error. Some structures of the network should be quantized last so the finetuning process can better adjust the centroids. We find that there are small variations in performance based on quantization order (see Figure 6). We choose to quantize FFN, then embeddings, and finally the attention matrices in Transformer networks.

**Which structures can be compressed the most?** Finally, we analyze which network structures can be most compressed. During quantization, various matrix block sizes can be chosen as a parameter — the larger the block size, the more compression, but also the larger the potential reduction of performance. Thus, it is important to understand how much each network structure can be compressed to reduce the memory footprint of the final model as much as possible. In Figure 6, we quantize two model structures with a fixed block size and vary the block size of the third between 4 and 32. As shown, the FFN and embedding structures are more robust to aggressive compression, while the attention drastically loses performance as larger block sizes are used.

### 7.11.5 APPROACH TO `int`N SCALAR QUANTIZATION

We compare quantizing per-channel to using a histogram quantizer in Table 10. The histogram quantizer maintains a running min/max and minimizes L2 distance between quantized and non-quantized values to find the optimal min/max. Quantizing per channel learns scales and offsets as vectors along the channel dimension, which provides more flexibility since scales and offsets can be different.

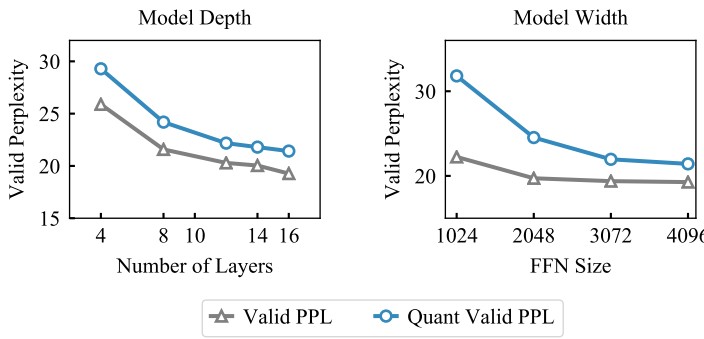

Figure 5: **(a)** Effect of Initial Model Size for more shallow models **(b)** Effect of Initial Model Size more skinny models

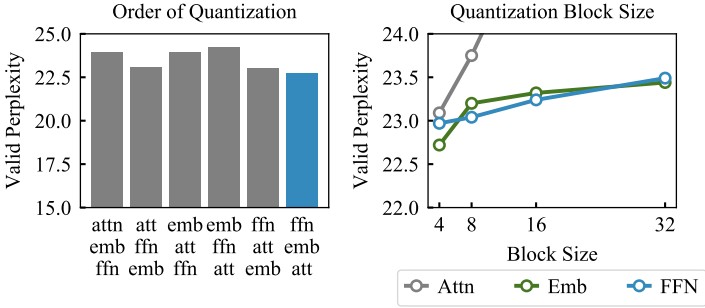

Figure 6: **Effect of Quantization on Model Structures.** Results are shown on the validation set of Wikitext-103. **(a)** Quantizing Attention, FFN, and Embeddings in different order. **(b)** More Extreme compression of different structures.

### 7.11.6 LAYERDROP WITH STE

For quantization noise, we apply the straight through estimator (STE) to remaining weights in the backward pass. We experiment with applying STE to the backward pass of LayerDrop's pruning noise. Results are shown in Table 11 and find slightly worse results.

| Quantization Scheme | Language Modeling 16-layer Transformer Wikitext-103 | | | Image Classification EfficientNet-B3 ImageNet-1K | | |
|---|---|---|---|---|---|---|
| | Size | Compress | Test PPL | Size | Compress | Top-1 Acc. |
| Uncompressed model | 942 | ×1 | 18.3 | 46.7 | ×1 | 81.5 |
| Int4 Quant Histogram | 118 | ×8 | 39.4 | 5.8 | ×8 | 45.3 |
| + Quant-Noise | 118 | ×8 | 21.8 | 5.8 | ×8 | 67.8 |
| Int4 Quant Channel | 118 | ×8 | 21.2 | 5.8 | ×8 | 68.2 |
| + Quant-Noise | 118 | ×8 | 19.5 | 5.8 | ×8 | 72.3 |
| Int8 Quant Histogram | 236 | ×4 | 19.6 | 11.7 | ×4 | 80.7 |
| + Quant-Noise | 236 | ×4 | 18.7 | 11.7 | ×4 | 80.9 |
| Int8 Quant Channel | 236 | ×4 | 18.5 | 11.7 | ×4 | 81.1 |
| + Quant-Noise | 236 | ×4 | 18.3 | 11.7 | ×4 | 81.2 |

Table 10: **Comparison of different approaches to `int4` and `int8` with and without Quant-Noise** on language modeling and image classification. For language modeling, we train a Transformer on the Wikitext-103 benchmark. We report perplexity (PPL) on the test set. For image classification, we train a EfficientNet-B3 on the ImageNet-1K benchmark. We report top-1 accuracy on the validation set. For both setting, we also report model size in megabyte (MB) and the compression ratio compared to the original model.

| Model | MB | PPL |
|---|---|---|
| Quant-Noise + Share + Prune | 10 | 24.2 |
| Quant-Noise + Share + Prune with STE | 10 | 24.5 |

Table 11: **Performance on Wikitext-103 when using STE in the backward pass of the Layer-Drop pruning noise.**

