# OpenReview forum: "Training with Quantization Noise for Extreme Model Compression"
_ICLR.cc/2021/Conference — ICLR 2021 Poster_

### Official Review · AnonReviewer2 · 2020-10-27

**Rating:** 4
**Confidence:** 4

**Review:**

In this paper, the authors propose a new compression-aware training method utilizing quantization noise as a regularization technique. During training, only a different random subset of weights is quantized for forward propagation such that controlling the amount of noise is a way to improve model accuracy when extreme compression is applied.

Even though the motivation is interesting and the selection of recently developed models (including EfficientNet-B3 and RoBERTa) is reasonable, this reviewer has the following critical concerns:

- It is hard to understand what the paper aims to describe because several equations in the paper seem to be very confusing or incorrect.
1) Since $z$ is an integer in equation (2) (i.e. $z$ is a rounded value), $(round(W/s+z)-z)*s$ is equal to $c=round(W/s)*s$. Then, it seems that we just round the real-valued weights with only a scaling factor without zero-point. This reviewer is wondering whether $z$ means 'zero-point'. It is very confusing.
2) In equation (5), as far as I understand, $n$ seems to be the term related to activation, but there is no explanation about $n$.
3) The term in equation (8), $y_{noise} * x^T$, needs to be explained in the aspect of gradients. If we follow equation (8), then we may update the weights with only the product of input and output of a layer. This reviewer cannot understand what the authors mean by equation (8). To understand the training process of the proposed quant-noise method, equation (8) seems to be important. However, there is no explanation for how it was derived.
4) Some scalar values (e.g. $W_{kl}$ or $I_{kl}$) are denoted by using matrix notations.

- There is no enough analysis on why quant noise can improve the quantization accuracy. The training process can be affected by various hyperparameters such as learning rates, dropout prob., weight decay, and so on. The authors are encouraged to present how regularization-related hyper-parameters are affected and to clarify the effects of the quant-noise method on the model accuracy. The current manuscript lacks such an analysis. In addition, there are no graphs showing the training process with the proposed quant-noise method. The training graph with various learning rates or regularization parameters could have elaborated on the contributions of this paper.

- The comparisons in Table 1 are not fair and detailed analysis and discussion of the experimental results are required. The authors show poor results trained with QAT and claim that the proposed method can improve the accuracy significantly. But, this reviewer cannot understand why the quantization-aware training method is not better than the post-processing method (i.e., intN quantization). Also, it should be clarified whether int4 quantization and int8 quantization are post-processing quantization or not.
When training with QAT, it will be affected by many hyper-parameters and there may be a decrease in performance (PPL and Top-1). It should be discussed that the QAT results are well-trained results.

- Moreover, it is difficult to compare the effects of quant-noise because there have been no previous works that have performed quantization in the models described in the manuscript. The authors select three models to show the effects of the method those are Transformer LM, EfficientNet-B3 (why not B1/B2?), and Roberta. But, the rather well-known models (e.g., ResNet, MobileNet, BERT, and so on) could be better than the selected models in the manuscript. For example, a highly related paper, iPQ paper (Stock et al., 2019), used the ResNet-18 and 50 models. This reviewer is wondering why the authors do not choose the models for easy and fair comparisons.



In summary, this reviewer cannot estimate the advantages of the proposed technique because the training process is not described in detail and the experimental results are not fair.


 <Minor comments>

- In Table 3, I think the specific model name should come instead of "adaptive input".

- typos
 3page:
     (weights) simulatneously -> simultaneously
     (Bayesian) intepretation -> interpretation
 4page:    (and) learns -> learn
 8page:     (These) comparison -> comparisons

---

> ### Author Response · Authors · 2020-11-19
> **Answer**
>
> re: equations
> - Regarding Equation (2), it performs “fake quantization” in the sense that it successively quantizes and dequantizes the weights. Therefore, the reviewer is right that when successively quantizing and de-quantizing the weights, the zero point or bias $z$ cancels out (which is normal, see [1]). However, we found this equation interesting as it shows both the quantization operation $\text{round}(\mathbf W_{kl}/s + z)$ (where the zero point plays a role) and its inverse dequantization operation. We will modify the paper to better reflect the two operations.
> - Regarding Equation (5), recall that $\mathbf W \in \mathbb R^{n \times p}$ in Equation (1), therefore, with batch size 1, the input activations have size $1 \times n$, which explains the term $8 \times n$ in Equation (5). For more clarity, we will modify the paper to recall the meaning of $n$ and we thank the reviewer for their feedback.
> - Regarding Equation (8), using back-propagation to compute the gradient of the loss $\mathcal L$ with respect to $\mathbf W$, we obtain $\frac{\partial L}{\partial y}\mathbf x^T$ (and not $\mathbf y \mathbf x^T$ as indicated in the manuscript). Note that here, $\frac{\partial L}{\partial y}$ is obtained by replacing the weights by their version *without* noise. We thank the reviewer for pointing this out and hope this clarifies the use of the STE. We will modify the paper accordingly.
> - Regarding the scalar values, $\mathbf b_{kl}$ is in general a vector (or block) as indicated in Equation (1). Therefore we propose to keep the notation unchanged, even though in some cases (m=1 and q=1), it may default to a scalar.
>
>
> re: analysis of model parameters - Quant-Noise adds one parameter, the amount of quantization noise added at training time. This parameter affects the amount of the network that is pseudo-quantized during training. All other parameters are parameters of the quantization method, not of ours. Analysis of how different levels of quantization noise affect the network as other parameters vary is presented in the Appendix already, see Appendix Section 7 and Figure 4, Figure 5, and Figure 6. Regarding the effect of regularization parameters with quantization noise, we found that Quant-Noise has a regularization effect, so other regularization parameters can be reduced.  Finally, all model parameters are presented in Appendix Section 7 in detail. If there are questions about specific parameters, we can easily add ablations, but believe that the Appendix provides detailed investigation already.
>
> re: Table 1 is not fair - We perform QAT by tuning the QAT parameters to get a strong QAT baseline. However, note that training with QAT actually performs the quantization during training time fully, rather than only applying proxy Quant-Noise, like our proposed method. QAT introduces a discrepancy between training and testing, which we discuss in the introduction, because of the use of STE to handle the gradients. This is why QAT performs well in certain settings, but not well in all settings.
>
> re: Why not quantize different models?   We wanted to quantize the most difficult and state of the art models in this paper. ResNet and MobileNet, for example, are easier to quantize than EfficientNet, because EfficientNet is a smaller model. Larger models are more redundant and thus easier to achieve high compression ratios. Regarding BERT, actually RoBERTa is a better version of BERT.  We chose the state of the art models in vision and NLP as testbeds for our technique to show that it can aggressively compress networks while retaining strong performance.
>
> New Result on ResNet: In all our experiments with different architecture, Quant-Noise systematically achieves better compression ratios compared to Stock et al on ImageNet. For instance, if we compress a ResNet-50 (uncompressed top-1 ImageNet accuracy 76.15%) with Quant-Noise, we improve the performance for both the small blocks and large blocks regime (Stock et al.). To fairly compare accuracy, we hold the compression rate fixed to the same amount as Stock et al. We include the table below and have added it to the paper. As shown in the table, in both settings we improve upon the result from Stock et al.
>
> |Compression | Model | Top-1 Accuracy | Compression Rate |
> |---|---|---|---|
> |None | Uncompressed | 76.2% | 1x |
> | Small blocks | Stock et al. | 73.8% | 19x|
> | Small blocks | Quant-Noise | 74.3% | 19x|
> | Large blocks | Stock et al. | 68.2%|31x|
> | Large blocks | Quant-Noise | 68.8%|31x|
>
> re: typos - Thanks! We will fix all the typos and appreciate the detailed read of our paper.
>
> References:
> [1] https://github.com/pytorch/pytorch/blob/master/torch/quantization/fake_quantize.py#L50

---

### Official Review · AnonReviewer4 · 2020-10-27
**The paper presents a new technique to quantize the weights and the activations of neural network models during the training. The authors first introduce Quant-noise, a method to simulate quantization during the training. Then Quant-Noise is applied to quantization methods such as int8 and Product Quantization, which are combined to obtain the final results.**

**Rating:** 10
**Confidence:** 5

**Review:**

Reasons for Score:
The work proposed in this paper is novel and very well presented. The claims are supported by experimental results on different neural network models and applications, showing a good trade-off between accuracy and neural network compression.

Strengths:
1.	The Quant-Noise idea for quantizing the weights and activations of neural networks is innovative, and it can be easily applied to various quantization methods for compressing the neural networks and saving energy.
2.	The results are very good in terms of compression rate and accuracy, and the experiments are obtained on complex state-of-the-art neural network models and applications.
3.	The combination of int8 and Product Quantization results in hardware-deployable compressed models.
4.	The background and the methods used are clearly explained, also from the mathematical and formal point of view.
5.	The paper is overall clear and well written.

Minor Comments:
1.	Which is the training time overhead of having quantization noise during training?
2.	The percentage (%) should be indicated in the vertical axes of Figure 2.

---

> ### Author Response · Authors · 2020-11-19
> **Answer**
>
> We thank the Reviewer for showing interest in our work. We argue indeed that Quant-Noise is a novel, simple idea for quantizing neural networks that works on a variety of architectures, tasks and quantization methods (scalar or product quantization), allowing to reach a good compromise between model size and accuracy.
>
> Regarding the training time overhead: when training the uncompressed network, Quant-Noise masks out some blocks of the weights. This is equivalent to point-wise multiplying every weight matrix during the forward pass on-the-fly with a mask of zeros and ones (we did not optimize this part for speed but rather for readability). In our experiments, training time was less than 5% higher than training without Quant-Noise, which amounts in practice to adding 1 or 2 hours of training time. Note that the backward pass timing with STE is unchanged since we keep the non-distorted matrix in memory. We will clarify this point in the paper.
>
> Thanks for pointing out the missing percentage, we will update the paper accordingly.

---

### Official Review · AnonReviewer3 · 2020-10-28
**This work proposes a simple modification to the STE-based quantization aware training to improve the accuracy of quantized neural networks. This proposed method achieves significant accuracy improvement when using INT4 quantization on ImageNet and Wikitext. The introduced quant-noise function is flexible as it can combine pruning with quantization to further reduce the size of the network size.**

**Rating:** 6
**Confidence:** 3

**Review:**

I agree with the three key contributions listed in the paper. The paper is well written and clearly articulates a contribution to the literature. The proposed Quant-Noise is intuitive and straightforward. The experimental evidence is provided for both image classification and language modeling tasks. Most of the related works are cited. The paper does not contain a theory part, but wherever possible, equations are provided to illustrate how the method works.

Concerns:
For the experimental results, only one network is used to evaluate the proposed method. With different architecture, the accuracy gap between the full precision and quantized models could be different. It will be helpful to validate the proposed Quant-Noise on multiple networks. Moreover, the experimental evidence only shows results for INT4, INT8, and iPQ. As the proposed method should also work for fixed-point quantization and even binarization. The effectiveness of the proposed approach can be better justified by comparing the results with other SOTA fix-point quantization and binarization schemes in [1-4].

I also have a question about the selection of the weights. The current scheme selects a random subset of the weights to add the proposed noise, which means the weights in different layers are chosen with the same probability. For pruning, existing work [5] finds that it is desired to prune different fractions of weights for each layer. Is there a better way to select the subset of the weights?

Reasons for score: Overall, I am leaning towards accepting the paper. I like the simple idea of controlling the imprecise gradients in BNN training by only quantizing a subset of weights at each iteration. However, fewer comparisons are made with other SOTA quantization/binarization methods. I would consider raising my score if the authors could address the aforementioned concerns.

[1] Bi-Real Net: Enhancing the Performance of 1-bit CNNs With Improved Representational Capability and Advanced Training Algorithm

[2] Searching for Low-Bit Weights in Quantized Neural Networks

[3] Differentiable Soft Quantization: Bridging Full-Precision and Low-Bit Neural Networks

[4] LQ-Nets: Learned Quantization for Highly Accurate and Compact Deep Neural Networks

[5] Pruning neural networks without any data by iteratively conserving synaptic flow

---

> ### Author Response · Authors · 2020-11-19
> **Answer**
>
> We thank the reviewer for agreeing with the contributions listed in the paper, in particular that the authors provide empirical evidence showing that the proposed method works.
>
> Regarding the experimental results:
> We chose to push for performance by targeting state-of-the-art architectures -- namely EfficientNets for vision and Transformers for NLP -- instead of exploring more model variants. Overall, we evaluate our method for both scalar quantization (int4 and int8) and product quantization, on three tasks (image classification, language modeling and RoBERTa on MNLI). To showcase the generality of our method, we compress a ResNet-50 (uncompressed top-1 ImageNet accuracy 76.15%) with Quant-Noise and improve the performance for both the small blocks and large blocks regimes (Stock et al.) by 0.5-0.6% in absolute top-1 accuracy, to respectively 74.3% and 68.8%, see below table:
>
>
>
> |Compression | Model | Top-1 Accuracy | Compression Rate |
> |---|---|---|---|
> |None | Uncompressed | 76.2% | 1x |
> | Small blocks | Stock et al. | 73.8% | 19x|
> | Small blocks | Quant-Noise | 74.3% | 19x|
> | Large blocks | Stock et al. | 68.2%|31x|
> | Large blocks | Quant-Noise | 68.8%|31x|
>
> Although investigating lower-bit quantization is an interesting direction (as pointed out by the Reviewer), we argue that (1) when exploring scalar binarization (1 bit per weight), Product Quantization offers a better tradeoff and should be considered instead and (2) dedicated hardware is not widely used and/or available for scalar quantization lower than int4.
>
> Regarding the selection of the weights: this direction is indeed an interesting one to explore. Here are some early preliminary experiments (not included in the paper) that we conducted. For EfficientNets in the Product Quantization setup, we observed that the 1x1 convolutions were harder to quantize than the 3x3 convolutions (in terms of observed drop in accuracy). We applied more noise to the 1x1 convolutions during training with Quant-Noise, which resulted in a lower uncompressed training accuracy. We then observed that the layer-wise quantization error (in terms of L2 norm) was *lower* with this model, hence that the quantization was performing a bit better. However, the two effects (lower uncompressed training accuracy, better quantization) were offsetting and we ended up with a slightly worse performing compressed model. However, there may be some setups in which this idea could help. We thank the reviewer for this interesting suggestion.

---

### Official Review · AnonReviewer1 · 2020-10-29
**Nice Results but More Analysis is Required.**

**Rating:** 4
**Confidence:** 4

**Review:**

Contribution:
1. conduct Quantization-Aware-Training via introducing quantization noise during optimization. Further, Quant-Noise can be applied to existing trained networks
2. The proposed method is validated on multiple models like Transformer, ConvNet, and more challenging EfficientNet-B3. Experiential results show that the proposed method is practical.


Cons:
Product Quantization (PQ) is not of much novelty since it was studied by Stock et al. This paper combines fixed-point and PQ and tries to claim that 'Fixed-point quantization and Product Quantization are often regarded as competing choices'. However, this claim is under-explained.


One of important novel points in this paper is subset quantization. A potential advantage of subset quantization is to mitigate the Gradient Mismatch problem caused by STE. This motivation makes sense to me because the gradient of unselected parameters is unbiased. This unbiased gradient will improve the quality of the gradient for the previous layers. However, a similar training scheme for both pruning and quantization was studied in related work [1, 2].

[1] Guo, Yiwen, Anbang Yao, and Yurong Chen. "Dynamic network surgery for efficient dnns." Advances in neural information processing systems. 2016.
[2] Zhou, Aojun, et al. "Incremental network quantization: Towards lossless cnns with low-precision weights." arXiv preprint arXiv:1702.03044 (2017).

There are some technical differences among these methods. However, considering such a similarity, more analysis is necessary to help us understand why subset quantization benefits (and helps reduce gradient mismatch). Otherwise, the novelty of this submission will be incremental.

---

> ### Author Response · Authors · 2020-11-19
> **Answer**
>
> Thanks for your review. We argue indeed that Quant-Noise is a novel, simple idea for quantizing neural networks that works on a variety of architectures, tasks and quantization methods (scalar or product quantization), allowing to reach a good compromise between model size and accuracy.
>
> Re: PQ is explored by Stock et al - Quant-noise is a general technique that can be applied easily to any quantization method, whether Product Quantization or Scalar Quantization. We do not claim Product Quantization as a novelty, but much stronger performance due to Quant-Noise. You can see this in Table 1, where we compare PQ from Stock et al to PQ + Quant-Noise. On Language Modeling, adding Quant-Noise improves over 4 PPL points. For ImageNet, adding Quant-Noise improves 1% absolute.
>
> Re: Fixed-point and PQ are regarded as competing choices - We will clarify this in the final version of the paper. We mean that previous work using PQ (such as Stock et al) does not combine PQ with Scalar Quantization methods like int8 quantization. Our work shows that Quant-Noise is effective at improving the performance and compression of each technique separately, as well as both together.
>
> Re: Why subset quantization helps - We will add a citation to the related work. The main improvement from training with Quant-Noise comes from the network being exposed to the quantization method during training time. However, if the network were fully quantized during training, such as using STE, a discrepancy would be introduced between the training and testing schemes. This is ok for small compression ratios (like int8), but works poorly for extreme compression, which we explore in our work.
>
> There are two main advantages to Quant-noise: (1) Much stronger performance at large compression ratios compared to previous work. See Figure 2, where we compare the resulting performance and model size across three tasks and two domains to previously published works. We show gains over Quantization-Aware-Training (Table 1) as well. (2) Straightforward to apply. Quant-Noise can be added in a few lines of code to existing quantization techniques.

---

### Official Review · AnonReviewer5 · 2020-11-06
**Official Blind Review #5**

**Rating:** 5
**Confidence:** 4

**Review:**

This paper introduces Quant-Noise that quantizes a random fraction of the network at each step instead of quantizing the entire network. The experiments show that the proposed Quant-Noise can improve the accuracy of quantized neural networks.

Pros:
1. The proposed technique is simple and easy to use.
2. The proposed Quant-Noise shows very impressive accuracy improvements for int4 quantization.

Cons:
1. The novelty of the proposed method is limited. Especially when combined with product quantization, Quant-Noise is essentially a form of structured dropout. I agree adding structured dropout may lead to some accuracy improvements. But I do not think this is one of the contributions of this paper. Besides, why Quant-Noise works for int4 quantization is not clear.
2. The improvement of Quant-Noise for int8 quantization looks very limited. The proposed method seems to be useful for low-bit quantization, but not very effective for int8 quantization. If so, I think the authors should focus on low-bit quantization in the experiment section and test the proposed method on more low-bit quantization settings  (e.g., 2bits quantization, 3bits quantization, mix-precision quantization, etc).

Overall, I think the novelty of the proposed method is limited and why it works is not very clear. Besides, I think the proposed method should be tested on more low-bit quantization settings.  I recommend "Marginally below acceptance threshold".

---

> ### Author Response · Authors · 2020-11-19
> **Answer**
>
> We thank the reviewer for underlying the simplicity of the technique and its ability to provide accuracy improvements. We address the comments below.
>
> Regarding the novelty of the method: we argue that using structured dropout with the specific goal of improving post-training quantization is novel. Moreover, Quant-Noise works for both scalar and vector quantization since it helps the network adapt to future quantization operations during the training. More generally, we generalize the notion of dropout to more general quantization noise with a purpose: the one of better aligning the training with the final quantization in a way that keeps gradients flooding. We believe this also explains why this works for int4, for which the noise structure is different from more standard dropout.
>
> For int8 quantization, the limited gain in performance comes from the fact that int8 quantization is not an aggressive quantization method, therefore the expected gains are smaller than with Product Quantization for instance because the loss of accuracy due to the lack of precision is limited in the first place. That's being said, even a small gain can make a difference in some applications and our method with int8 is straightforward to use, therefore we believe the practitioners may consider it in that context.
>
> Although investigating lower-bit quantization is an interesting direction (as pointed out by the Reviewer and also by R3), we argue that (1) when exploring scalar binarization (1 bit per weight), Product Quantization offers a better tradeoff and should be considered instead and (2) dedicated hardware is not widely used for scalar quantization lower than int4.

---

### Author Response · Authors · 2020-11-19
**Rebuttal Revision**

Dear reviewers, thanks again for your improvement suggestions and questions. We have updated our manuscript to address the raised concerns, including:
- Results on ResNets
- Context/explanations for equations
- Typos

Thanks!

---

### Decision · Program_Chairs · 2021-01-07
**Final Decision**

**Decision:**

Accept (Poster)

**Comment:**

Quantization is an important practical problem to address.  The proposed method which quantizes a different random subset of weights during each forward is simple and interesting. The empirical results on RoBERTa and EfficientNet-B3 are good, in particular, for int4 quantization.  During the rebuttal, the authors further included quantization results on ResNet which were suggested by the reviewers. This additional experiment is important for comparing  this proposed approach with the existing methods which do not have quantization results on the models in this paper.